# Prevalence and Risk Factors of Obesity Among Type 2 Diabetic Participants in Abha, Saudi Arabia: A Cross-Sectional Study

**DOI:** 10.3390/healthcare13060658

**Published:** 2025-03-17

**Authors:** Fahad Abdullah Saeed Al Jarad, Bayapa Reddy Narapureddy, Hamza Radhwan Derkaoui, Abdulaziz Saud A. Aldayal, Meshari Mohammed H. Alotaibi, Faisal Hammad A. Aladhyani, Shaik Mohammed Asif, Kandasamy Muthugounder

**Affiliations:** 1Aseer Health Cluster, Ministry of Health, Abha 62312, Saudi Arabia; alshhranyfhd16@gmail.com; 2Department of Public Health, College of Applied Medical Sciences, Khamis Musait, King Khalid University, Abha 62525, Saudi Arabia; 3Public Health, Population Health Management, Aseer Health Cluster, Ministry of Health, Abha 62312, Saudi Arabia; derkamir@gmail.com; 4College of Medicine, Shaqra University, Shaqra 11911, Saudi Arabia; aldayelas@outlook.com (A.S.A.A.); m11msharee@gmail.com (M.M.H.A.); f.odhyani99@gmail.com (F.H.A.A.); 5Department of Diagnostic Science & Oral Biology, College of Dentistry, King Khalid University, Abha 61421, Saudi Arabia; masif020@gmail.com; 6Department of Community and Mental Health Nursing, College of Nursing, Khamis Mushait, King Khalid University, Abha 62525, Saudi Arabia; kandasamy.phd@gmail.com

**Keywords:** bio-demographic, type 2 diabetes mellitus, lifestyle factors, obesity, prevalence, risk factors

## Abstract

**Background:** The prevalence of obesity among type 2 diabetic participants is a growing concern globally, including in Abha, Saudi Arabia. This study aimed to investigate the prevalence and its associated risk factors of obesity among type 2 diabetic participants in Abha. **Methods:** A cross-sectional study targeted 400 type 2 diabetic participants in Abha, Saudi Arabia. A hybrid method (snowball sampling + purposive) techniques were used to obtain an adequate sample size. Data were collected after obtaining telephonic or digital consent; the questionnaire was shared with participants who were able and willing to complete it independently those with type 2 diabetes who wished to participate but were unable to complete the questionnaire on their own. The researcher conducted a telephonic interview and recorded their responses. The questionnaire captured demographic details, Anthropometric history, medical history, lifestyle habits, and type 2 Diabetes (T2DM) specific factors. Data were analyzed using IBM SPSS Statistics for Windows, Version 27.0. **Results:** The overall prevalence of obesity among the type 2 DM study participants was 46.0%, 115 participants (28.8%) fell into the Obesity Grade I category, 43 (10.8%) as Obesity Grade II, while 26 (6.5%) were classified as Obesity Grade III, the overall median BMI of participants was 29.3 ± 5.88. Significant bio-demographic factors associated with obesity included age, gender, educational level, marital status, and co-morbidities (*p* < 0.05). Notably, females and older adults exhibited higher obesity rates. Significant lifestyle factors included the frequency and type of physical activity, soft drink consumption, and attempts to control weight. Participants who exercised regularly and avoided soft drinks had lower obesity rates. Multiple logistic regression analysis identified age, gender, co-morbidities, family history of obesity, regular dinner consumption, soft drink consumption, and exercise frequency as significant predictors of obesity (*p* < 0.05). **Conclusions:** The study determined a high prevalence of obesity among type 2 diabetic participants in Abha, Saudi Arabia, with significant associations with bio-demographic and lifestyle factors. Interventions targeting weight management, physical activity, dietary habits, and health education are urgently needed to address obesity in this population. Further research is recommended to explore these associations longitudinally and to develop tailored intervention strategies.

## 1. Introduction

Obesity is a significant and modifiable risk factor for the development and progression of type 2 diabetes mellitus (T2DM) [1,2]. The rising prevalence of obesity globally has closely paralleled the increasing incidence of T2DM, underscoring the intricate relationship between excess weight and impaired glucose metabolism [3,4]. Excess body fat can alter the body’s metabolic system, promote insulin resistance, and trigger inflammatory responses that affect bodies’ blood sugar regulation. Excess adiposity contributes to insulin resistance by promoting chronic low-grade inflammation and lipid accumulation in non-adipose tissues, which disrupts insulin signaling and pancreatic β-cell function [5]. Recent studies suggest that up to 85% of adults diagnosed with T2DM are overweight or obese, further reinforcing obesity as a key driver of the disease [6]. A study in Dubai found that 49.5% of T2DM participants were obese, while 35.5% were overweight [7]. Similarly, research from Nigeria reported that 81.3% of individuals with T2DM were either overweight or obese at the time of diagnosis [8]. These findings highlight the critical need for targeted interventions to manage obesity as an essential component of diabetes prevention and treatment.

The etiology of obesity in T2DM is multifactorial, encompassing genetic, environmental, and behavioral factors. Genetic predisposition plays a significant role, as individuals with a family history of obesity and diabetes are at a higher risk of developing both conditions [9]. Lifestyle factors, particularly unhealthy dietary habits and sedentary behavior, substantially contribute to weight gain and obesity [10]. High-calorie diets rich in processed foods and sugary beverages exacerbate insulin resistance and dyslipidemia, both of which accelerate the onset of T2DM [11]. Physical inactivity compounds this risk, with evidence suggesting that a lack of exercise significantly increases the likelihood of obesity-related metabolic dysfunctions [12].

Socioeconomic and psychological factors also influence obesity prevalence among T2DM participants. Individuals from lower socioeconomic backgrounds often face barriers to accessing healthy food options and engaging in regular physical activity [13]. Moreover, stress, anxiety, and depression contribute to emotional eating and weight gain [14]. Certain medical conditions, including hypothyroidism and polycystic ovary syndrome (PCOS), are associated with an increased risk of obesity [15]. Additionally, pharmacological treatments for diabetes, such as insulin and some antipsychotic medications, can induce weight gain as a side effect, further complicating disease management [16,17].

In the United States, recent data indicate that over a third (34%) of adults are obese, with more than 11% aged 20 years or older diagnosed with diabetes [18]. In China, the prevalence of metabolic dysfunction-associated non-alcoholic fatty liver disease has risen with the increasing rates of obesity, metabolic syndrome, and T2DM [19]. These statistics highlight the critical need to address obesity in T2DM management. Although obesity is well-documented in T2DM participants, research gaps remain, particularly concerning its prevalence and risk factors in non-Western populations. Most studies primarily focus on Western cohorts, whereas limited data exist on obesity trends in Middle Eastern regions such as Saudi Arabia. This study aims to bridge that gap by evaluating the prevalence and determinants of obesity among T2DM participants in Abha, Saudi Arabia. Specifically, it will examine the impact of lifestyle factors, dietary habits, soft drink consumption, meal frequency, and socioeconomic influences on obesity. Addressing these risk factors necessitates a comprehensive approach that integrates lifestyle modifications, medical therapies, and support systems [20]. Evidence suggests that structured weight management programs including dietary interventions, regular exercise, and behavioral counseling can significantly improve glycemic control and reduce obesity-related complications in diabetic participants [21]. Pharmacological and surgical interventions, such as GLP-1 receptor agonists and bariatric surgery, may also be considered for individuals unable to achieve significant weight loss through conventional methods [22].

Given the strong association between obesity and T2DM progression, understanding its prevalence and risk factors is crucial for effective disease management and public health planning. This study aims to provide valuable insights into the interplay between obesity and diabetes in a Saudi Arabian context, supporting the development of tailored intervention strategies to mitigate obesity and its adverse metabolic consequences in T2DM participants.

## 2. Methodology

### 2.1. Study Design and Settings

This cross-sectional study was conducted to determine the prevalence of obesity and its risk factors associated with type 2 diabetes mellitus (T2DM). The study was conducted in Abha, Saudi Arabia from 10 November 2024 to 15 January 2025.

Study population: The target population included individuals aged 18 and above who have been diagnosed with type 2 diabetes and reside in Abha. Individuals diagnosed with type 2 diabetes and participants who provided informed consent to participate in the study were included. Non-residents of Abha, Saudi Arabia. Participants with other types of diabetes and those who did not provide informed consent were excluded.

### 2.2. Sample and Sampling

The minimum sample size was calculated based on the online caluculator.net sample size calculator. The sample size was calculated using the formula 4pq/d² at a 95% confidence interval level, where d is a 10% allowable error and the prevalence of obesity is 57.8% among Type 2 DM [23]. The minimum sample size calculated was 301, with a 10% non-response error of 33, so the total minimum required sample was 334. This study collected a total of 416 responses during the study period; of those, 14 responses were incomplete/non-resident in Abha and 2 responses were from individuals with Type 1 diabetes. After excluding those, a total of 400 Type 2 diabetic (T2DM) participants were included in the final analysis. A hybrid method (snowball sampling + purposive) techniques were used to obtain an adequate sample size.

### 2.3. Data Collection Tool and Techniques

Data collection was conducted following an extensive literature review of previous similar studies. A predesigned and pretested questionnaire was carefully developed to ensure its cultural appropriateness, reliability, and validity. To assess the validity and reliability of the questionnaire was piloted with 25 participants to refine the questionnaire, and enhance the clarity, effectiveness, and ease of understanding. Two independent public health experts thoroughly reviewed the questionnaire and provided inputs to modify some questions. Based on their inputs and insights gained from the pilot study, necessary adjustments and improvements were made to enhance its accuracy and comprehensibility. To determine the internal consistency reliability of the questionnaire, Cronbach’s alpha coefficient (0.78) was calculated, indicating an acceptable level of reliability. After finalizing the questionnaire, it was translated into Arabic by a professional translator and then back-translated into English by two independent Arabic experts to ensure that the meaning remained consistent and accurately conveyed.

The finalized questionnaire was deployed online using Google Forms, incorporating dropdown options to streamline responses and minimize the time required for participants to complete the survey. Due to data access restrictions, it is difficult to obtain a full detailed list of diabetic participants from health centers. The list of diabetic participants (restricted details) with their contact details like mobile numbers was obtained from local health centers. From this list, only participants with registered mobile numbers were contacted. After obtaining telephonic or digital consent, the questionnaire was shared with participants who were able and willing to complete it independently. The questionnaire was primarily distributed through WhatsApp and Snapchat, with participants, Participant-driven recruitment after filling out the survey questionnaire encouraged participants to forward it to other known literate type 2 diabetes individuals who are residing in the Abha area. For those with type 2 diabetes who wished to participate but were unable to complete the questionnaire on their own, the researcher conducted a telephonic interview and recorded their responses.

The questionnaire was designed to collect comprehensive data, in 5 sections including

Section 1: Demographic details: age, gender, educational level, marital status, work status, monthly income, and socioeconomic level.Section 2: Anthropometric history: weight (measured in kilograms with an accuracy of 0.5 kg) and height (measured in centimeters), recent measurements taken in clinical establishments of their last visit within the past 30 days from the survey date.Section 3: Medical history: duration of T2DM, family history of obesity, and co-morbidities such as hypertension and hyperlipidemia.Section 4: Lifestyle habits: physical activity levels, smoking status, and dietary patterns.Section 5: Diabetes-specific factors: efforts to control weight, soft drink consumption, and regular meal habits.

The questionnaire remained available online until the participant reached the desired sample size, no additional eligible participants were willing to participate, and no new responses were received.

### 2.4. Ethical Issues

Ethical approval was granted IRB Log No: REC-13-5-2024 dated 3 June 2024 by the Asser Institutional Review Board (IRB) of the Ministry of Health Directorate Health Affairs Asser region (H-06-B-091), and all participants were informed that their participation is completely voluntary nature they can withdraw from the study at any point of time without giving any explanation. The Consent was obtained in accordance with the declaration of Helsinki, through a digital agreement form.

### 2.5. Data Analysis

The data analysis for this study was conducted using IBM SPSS Statistics for Windows, Version 27.0 (IBM Corp., Armonk, NY, USA). Descriptive analysis was performed to summarize the socio-demographic characteristics, risk factors, and lifestyle habits of type 2 diabetic participants. Frequencies and percentages were calculated for categorical variables and the mean and standard deviation (SD) were computed for continuous variables like age. Inferential statistical analyses were applied to identify significant associations between obesity (as indicated by BMI) and various risk factors. Chi-square tests were used to determine the relationships between categorical variables and BMI categories, with a significance level of *p* < 0.05. An exact probability test was used for small frequency distributions.

## 3. Results

This study collected 416 total responses during the study period, in that 14 responses were incomplete/non-resident Abha and 2 responses were type 1 diabetes, after excluding those a total of 400 type 2 diabetic (T2DM) participants were included in the final analysis. The socio-demographic characteristics of 400 type 2 diabetic participants in Abha, Saudi Arabia. In terms of age distribution, 37 participants (9.3%) are under 40 years old, 94 (23.5%) are between 40 and 49 years, majority of the participants 117 participants (29.3%) are between 50 and 59 years, nearly one-quarter of the participants 95 (23.8%) are between 60 and 69 years, and 57 participants (14.3%) are 70 years or older, with a mean age of 55.1 years (SD = 12.4). The gender distribution is relatively balanced, with 207 males (51.7%) and 193 females (48.3%). Educational levels show that 104 participants (26.0%) are illiterate, 54 participants (13.5%) have basic education, 38 participants (9.5%) have secondary education, 44 participants (11.0%) hold a diploma, 70 participants (17.5%) are university graduates, and 90 participants (22.5%) have post-graduate education. Marital status indicates that 22 participants (5.5%) are single, 335 participants (83.8%) are married, and 43 participants (10.7%) are divorced or widowed. Regarding employment status, 187 participants (46.9%) were not working, 102 participants (25.6%) were non-healthcare staff, 11 participants (2.8%) were healthcare staff, and 99 participants (24.7%) were retired. Monthly income data reveal that 176 participants (44.0%) earn less than 5000 SR, 115 participants (28.8%) earn between 5000 and 10,000 SR, 88 participants (22.0%) earn between 10,000 and 15,000 SR, and 21 participants (5.2%) earn more than 15,000 SR. About socioeconomic levels, 19 participants (4.8%) are at a low level, 80 participants (20.0%) are below average, and 301 participants (75.2%) are at a satisfactory level. Co-morbidities among the participants show that 164 participants (41.0%) have hypertension, and an equal number have no co-morbid conditions. Additionally, 21 participants (5.3%) have hypothyroidism, 17 participants (4.3%) have asthma, 13 participants (3.3%) have hypercholesterolemia, and smaller percentages have other conditions such as heart disease (5 participants, 1.3%). Smoking status reveals that 352 participants (88.0%) are non-smokers, 13 participants (3.3%) are ex-smokers, and 35 participants (8.75%) are current smokers (Table 1).

Figure 1 illustrates the prevalence of obesity among type 2 diabetic participants in Abha, Saudi Arabia. Among the 400 participants surveyed, 74 individuals (18.5%) were classified as having normal weight. The majority of participants, 142 individuals (35.5%), were categorized as overweight. Furthermore, 115 participants (28.8%) fell into the Obesity Grade I category, indicating mild obesity. Obesity Grade II, or moderate obesity, was observed in 43 participants (10.8%), while 26 participants (6.5%) were classified as Obesity Grade III, corresponding to severe obesity. BMI ranged from 18.58 to 57.03 with a Median BMI of 29.3 ± 5.88.

Table 2 presents the distribution of type 2 diabetic participants’ BMI by bio-demographic factors. Age significantly influences BMI (*p* = 0.001), with the highest obesity rates found in participants aged 50–59 years (57.3%) and 60–69 years (56.8%). Gender also shows a significant association (*p* = 0.001), where 55.4% of females are obese compared to 36.9% of males. Educational level has a significant impact on BMI (*p* = 0.045), with higher obesity rates observed among illiterate participants (54.8%) and those with basic education (44.4%). Marital status is another significant factor (*p* = 0.010); divorced or widowed participants have a higher obesity rate (58.1%) compared to married (45.4%) and single individuals (31.8%). Co-morbidities significantly affect BMI (*p* = 0.001), with 52.1% of participants with co-morbid conditions being obese, compared to 37.2% of those without. Additionally, attempts to control or reduce weight are significantly related to BMI (*p* = 0.001), with 60.1% of participants who made such attempts being obese, compared to 35.2% of those who did not.

Table 3 illustrates the risk factors of obesity among type 2 diabetic participants. One significant factor is the duration of type 2 diabetes mellitus (DM) (*p* = 0.017). Participants with a duration of 5–9 years (53.6%) and those with a duration of less than 5 years (45.6%) showed higher obesity rates compared to other duration categories. Family history of obesity also shows a significant relationship with obesity (*p* = 0.019). Participants with both parents having a history of obesity are most affected, with 68.4% being obese, followed by those with one parent (54.2%) and none (41.6%). The consumption of soft drinks is another significant factor (*p* = 0.048). A higher percentage of participants who consume soft drinks (54.2%) are obese compared to those who do not (41.5%).

Table 4 illustrates lifestyle factors and their association with Body Mass Index (BMI) among type 2 diabetic participants. Firstly, the frequency of practicing exercises shows a significant relationship with BMI (*p* = 0.020). Participants who never exercise have a higher obesity rate (56.0%) compared to those who exercise irregularly (50.0%), once or twice a week (40.4%), many times a week (42.2%), or daily (35.4%). Secondly, the type of exercise also exhibits a significant relationship with BMI (*p* = 0.018). Notably, walking is the most common exercise, with 44.1% of participants who walk regularly being classified as obese compared to none of those who do strength and fitness exercises and weightlifting. Other types of exercise, such as running and team sports, show varying impacts on obesity rates.

Multiple stepwise logistic regression model for the risk factors of obesity among type 2 diabetic participants. Several significant factors were identified. Age is a significant factor (*p* = 0.042) with an adjusted odds ratio (ORA) of 1.02, indicating a slight increase in the likelihood of obesity as age increases. Female gender shows a strong association with obesity (*p* = 0.003) and has an ORA of 2.05, suggesting that females are more than twice as likely to be obese compared to males. Participants with co-morbidities also have a significantly higher risk of obesity (*p* = 0.048, ORA = 1.53). Interestingly, attempts to control weight are associated with a lower likelihood of obesity (*p* = 0.001, ORA = 0.26), indicating that efforts to manage weight can be effective. A family history of obesity is another significant factor (*p* = 0.028, ORA = 1.51), pointing to a genetic predisposition. Regular dinner consumption is associated with a lower risk of obesity (*p* = 0.035, ORA = 0.54), while the duration of type 2 DM shows a significant relationship (*p* = 0.049, ORA = 1.26), indicating that longer disease duration increases obesity risk. Regular consumption of soft drinks is significantly linked to higher obesity risk (*p* = 0.010, ORA = 1.95). Finally, regular exercise is associated with a reduced risk of obesity (*p* = 0.007, ORA = 0.79), highlighting the importance of physical activity in managing weight (Figure 2).

## 4. Discussion

The overall prevalence of obesity among type 2 diabetic participants in Abha, Saudi Arabia, was found to be 46.0%, with 28.5% classified as Obesity Grade I, 10.8% as Obesity Grade II, and 6.8% as Obesity Grade III. A study in Abha by Sachithananthan V noted that obesity grades prevalence in the diabetes mellitus subjects Grade I (31.1%) and Grade II (29.8%) and around 8.6% of the subjects had morbid obesity (≥40 BMI) (22). This is notably higher compared to the national prevalence of obesity in Saudi Arabia, which was reported to be 20.2% among adults in 2019 [23]. Another study in Bisha reported an even higher combined prevalence of overweight and obesity at 85.8%, with 27.9% of participants being overweight, 57.8% being obese, and only 13.2% having normal weight [24]. A study by Abdulsalam reported a high prevalence of overweight and obesity among type 2 diabetics (55.6%) [25].

Internationally, the prevalence of obesity has been increasing, with 16% of adults worldwide living with obesity in 2022 [26]. Similar findings to our study have been observed globally. In Tanzania, Damian et al. [27] reported a high prevalence of overweight and obesity among individuals with type 2 diabetes, reaching 85%. Furthermore, the prevalence was higher among females at 92.2%, compared to males at 69.9%. Similar trends have also been reported in the United States, Nepal, Ghana, and Iran [28,29,30,31,32]. The high prevalence of obesity among type 2 diabetic participants in Abha highlights the urgent need for targeted interventions to address this issue, as obesity is a significant risk factor for the development and progression of diabetes.

The study reveals that obesity is most prevalent among individuals aged 50–59 (57.3%) and 60–69 (56.8%), suggesting that metabolic decline and lifestyle changes in older age contribute to increased weight. This is consistent with findings from AlShhrani MS [23], who also reported higher obesity rates among older adults in Saudi Arabia, highlighting the increased risk of obesity with advancing age. Another significant discovery is the gender disparity in obesity prevalence, with females being over twice as likely to be obese compared to males (OR = 2.05, *p* = 0.003) [33]. These highlight potential sociocultural barriers that limit women’s engagement in physical activity. Gender also shows a significant association with BMI in our study, where females are more likely to be obese compared to males. Similar trends have been observed in other local studies, such as those by Algamdi M [34] and Al-Rubeaan et al. [35] which found that Saudi women had higher obesity rates than men, likely due to cultural and lifestyle factors that limit physical activity among women. Educational level also plays a crucial role, with higher obesity rates among illiterate individuals (54.8%) and those with only basic education (44.4%), compared to university graduates (34.3%). This aligns with global findings linking lower education levels to limited health awareness and access to preventive healthcare resources. This aligns with global findings, such as those from the World Health Organization (WHO), which indicate that lower educational attainment is associated with higher obesity rates due to limited access to health information and resources for healthy living [36]. While monthly income did not show a statistically significant correlation, individuals earning less than 5000 SR/month had the highest obesity prevalence (50.6%), indicating that financial constraints may limit access to healthier food and physical activity opportunities. The presence of co-morbidities was another major factor, with 52.1% of individuals with hypertension, hyperlipidemia, or other health conditions being obese, compared to 37.2% of those without co-morbidities. Co-morbidities significantly affect BMI, with participants having co-morbid conditions being more likely to be obese. This finding is supported by research from the American Diabetes Association (ADA), which reports that co-morbidities such as hypertension and dyslipidemia are common in obese diabetic participants, exacerbating their health risks [37]. This highlights the interconnected nature of metabolic disorders and the importance of a comprehensive healthcare approach for managing both diabetes and obesity. Lastly, individuals who actively attempted to control their weight had significantly lower odds of obesity (OR = 0.26, *p* = 0.001), suggesting that weight management strategies can be effective in this population. Attempts to control or reduce weight show a significant relationship with BMI, indicating the effectiveness of weight management efforts. This is reflected in global strategies by organizations such as the Centers for Disease Control and Prevention (CDC), which promote comprehensive weight management programs customized to individual needs [38]. Furthermore, efforts to manage weight, such as regular physical activity and dietary modifications, have been shown to significantly improve BMI and contribute to better diabetes management [38]. These findings underscore the complex interplay of bio-demographic factors influencing obesity among type 2 diabetic participants.

Other factors also showed a significant relation with obesity among type 2 diabetics. Duration of type 2 diabetes mellitus (DM) is a notable factor, with participants having diabetes for 5–9 years or fewer than 5 years showing higher obesity rates. This correlation between diabetes duration and obesity is consistent with findings from global studies, such as those by the International Diabetes Federation (IDF), which indicate that longer diabetes duration often correlates with higher obesity rates due to prolonged exposure to insulin resistance and lifestyle changes [39]. A significant relationship was also observed between a family history of obesity and current obesity rates, with the highest obesity prevalence among participants whose parents had a history of obesity. Additionally, the consumption of sugar-sweetened beverages has been strongly linked to weight gain and obesity. Overconsumption of these drinks is associated with obesity, hypertension, type 2 diabetes, dental caries, and low nutrient levels. Experimental studies have reported that sugar-sweetened soft drinks potentially contribute to these ailments, though some studies show conflicting information. A 2013 systematic review found that 83.3% of reviews without reported conflicts of interest concluded that sugar-sweetened soft drink consumption could be a potential risk factor for weight gain Calcaterra V et al. [40], reported that sugary beverage consumption is strongly linked to weight gain and obesity due to high-calorie and sugar content.

The analysis of lifestyle factors and their association with Body Mass Index (BMI) among type 2 diabetic participants reveals significant relationships. Firstly, the frequency of exercise shows a notable correlation with BMI. Participants who never exercise have the highest obesity rates, which decreases progressively with increased frequency of exercise. Recent studies have highlighted significant associations between lifestyle factors and Body Mass Index (BMI) among individuals with type 2 diabetes. This aligns with findings from Jayedi et al. [41], who conducted a systematic review and meta-analysis showing that increased physical activity significantly reduces the risk of obesity and type 2 diabetes. Furthermore, the American College of Sports Medicine [42] recommends a minimum of 150 min of moderate-intensity aerobic exercise per week for effective weight management and diabetes control. In addition to exercise frequency, the type and intensity of physical activity also play a crucial role in managing BMI. Walking, being a common form of exercise, is associated with a relatively higher prevalence of obesity compared to more intensive activities such as strength training and high-intensity interval training (HIIT). Research by Sardar et al. [43] indicates that high-intensity exercises are more effective in reducing body fat percentage and improving cardiovascular fitness among individuals with obesity. Similarly, Colberg et al. [44] emphasize that engaging in resistance training and aerobic exercises can improve insulin sensitivity and overall metabolic health in diabetic participants.

This study presents several novel findings regarding obesity risk factors among type 2 diabetic (T2DM) participants in Abha, Saudi Arabia. One significant discovery is the gender disparity in obesity prevalence, with females being over twice as likely to be obese compared to males (OR = 2.05, *p* = 0.003). This highlights potential sociocultural barriers that limit women’s engagement in physical activity [22,34]. Another finding is that individuals who actively attempted to control their weight had significantly lower odds of obesity (OR = 0.26, *p* = 0.001), Menghui Liu et al. suggesting that weight management strategies can be effective in the population [45]. Interestingly, regular dinner consumption was associated with a lower risk of obesity (OR = 0.54, *p* = 0.035), contradicting traditional beliefs that skipping meals aids weight loss (9). Additionally, frequent soft drink consumption showed a strong association with obesity (OR = 1.95, *p* = 0.010), reinforcing concerns about the role of sugar-sweetened beverages in weight gain [40]. These findings emphasize the need for culturally tailored lifestyle interventions, particularly in dietary habits and physical activity.

The obesity prevalence among T2DM participants in Abha (46.0%) is more than twice the reported national average of 20.2%, suggesting that diabetic individuals are significantly more vulnerable to obesity [22]. This highlights the urgent need for targeted obesity prevention efforts among this high-risk group. The study also reinforces the link between soft drink consumption and obesity, a pattern observed globally. Including a comparative analysis with non-diabetic individuals could further clarify how lifestyle behaviors differ between diabetic and non-diabetic populations, strengthening arguments for targeted public health interventions [40].

The study’s findings have significant clinical and public health implications for obesity management among diabetic participants in Saudi Arabia. From a clinical perspective, healthcare providers should prioritize personalized weight management strategies, particularly for women and older adults who exhibit higher obesity risks. Screening for a family history of obesity should also be integrated into routine healthcare, given its strong association with obesity risk (OR = 1.51, *p* = 0.028) [9]. Additionally, healthcare professionals should encourage regular physical activity, as regular exercise was found to significantly reduce the likelihood of obesity (OR = 0.79, *p* = 0.007) [45]. From a public health standpoint, targeted interventions should focus on reducing soft drink consumption, promoting structured exercise programs, and improving dietary education. Gender-specific community programs may help address sociocultural barriers that limit physical activity for women. Moreover, nutritional education campaigns should challenge misconceptions about meal skipping, especially since the study found that regular dinner consumption was associated with a lower obesity risk [22].

### 4.1. Limitations

When interpreting the results of this study, several limitations should be noted. The cross-sectional design prevents establishing causality between identified risk factors and obesity, necessitating further longitudinal studies. The reliance on self-reported Measurement errors in weight and height may have influenced BMI calculations and may introduce bias. Limit generalizability due to snowball sampling. Importantly this study did not include a comparison group of non-diabetic participants to understand the BMI differences as contributing factors for DM progression could have been achieved. Additionally, the study did not consider confounding factors such as medication use and genetic predisposition. Despite these limitations, the study provides valuable insights into obesity prevalence and risk factors among type 2 diabetic participants, suggesting areas for further research and intervention.

### 4.2. Conclusions and Recommendations

This study highlights the high prevalence of obesity among T2DM participants in Abha, Saudi Arabia, with strong associations with sedentary lifestyles, poor dietary habits, and socioeconomic factors. Key contributors include high sugar consumption, low physical activity, and lower education levels. Notably, gender disparities suggest the need for targeted health interventions. Addressing obesity in this population requires structured weight management programs, emphasizing dietary modifications and physical activity promotion within a culturally appropriate framework. Given the study’s reliance on self-reported data and non-probabilistic sampling, future research should incorporate clinical measurements and randomized samples to enhance generalizability. Policy-driven strategies are essential to mitigate obesity-related complications and improve diabetes management.

## Figures and Tables

**Figure 1 healthcare-13-00658-f001:**
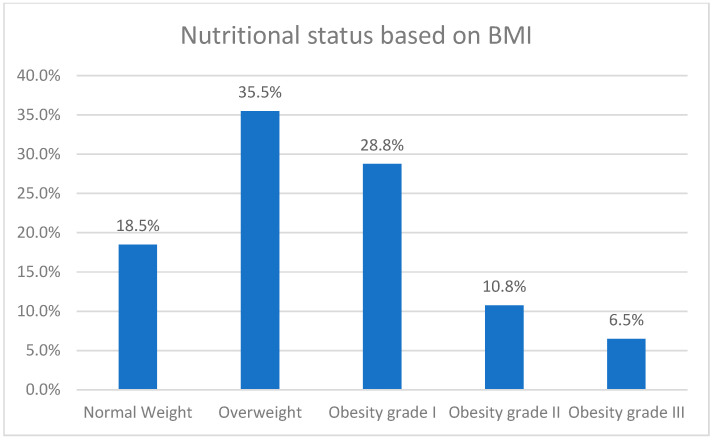
Prevalence of obesity among type 2 diabetic participants in Abha, Saudi Arabia (n = 400).

**Figure 2 healthcare-13-00658-f002:**
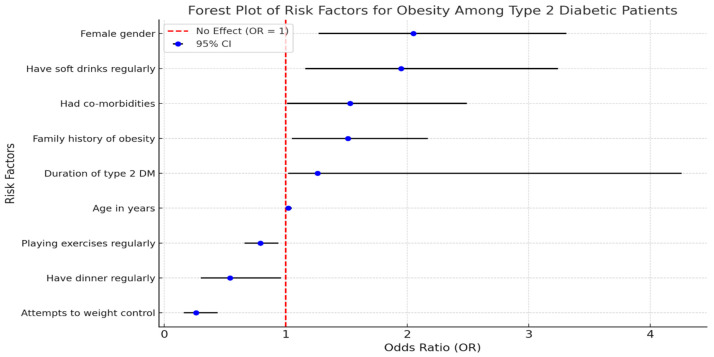
Forest Plot of Risk Factors for Obesity Among Type 2 Diabetic Participants.

**Table 1 healthcare-13-00658-t001:** Socio-demographic characteristics of the study type 2 diabetic participants, Abha, Saudi Arabia (n = 400).

Bio-Demographic Data	No	
Age in years		
<40	37	9.25%
40–49	94	23.50%
50–59	117	29.25%
60–69	95	23.75%
70+	57	14.25%
Means ± SD	55.1 ± 12.4	
Gender		
Male	207	51.75%
Female	193	48.25%
Educational level		
Illiterate	104	26.00%
Basic education	54	13.50%
Secondary education	38	9.50%
Diploma	44	11.00%
University graduate	70	17.50%
Post-graduate	90	22.50%
Marital status		
Single	22	5.50%
Married	335	83.75%
Divorced/widow	43	10.75%
Work status		
Not working	188	47.00%
Non-health care staff	102	25.50%
Health care staff	11	2.75%
Retired	99	24.75%
Monthly income		
<5000 SR	176	44.00%
5001–10,000 SR	115	28.75%
10,001–15,000 SR	88	22.00%
>15,001 SR	21	5.25%
Socioeconomic level		
Low level	19	4.75%
Below average	80	20.00%
Satisfactory	301	75.25%
Co-morbidities		
None	164	41.00%
HTN	164	41.00%
hypothyroidism	21	5.25%
Asthma	17	4.25%
hypercholesteremia	13	3.25%
Others	10	2.50%
hypocholesteremia	6	1.50%
Heart disease	5	1.25%
Smoking		
Non-smoker	352	88.00%
Ex-smoker	13	3.25%
Current smoker	35	8.75%

**Table 2 healthcare-13-00658-t002:** Distribution of type 2 diabetic participants' BMI by their bio-demographics.

Bio-Demographic Factors	Body Mass Index								*p*-Value
	Normal Weight		Overweight		Obese		Total		
	No	%	No	%	No	%	No	%	
Age in years									0.001
<40	12	32.4	13	35.1	12	32.4	37	100	
40–49	22	23.4	39	41.5	33	35.1	94	100	
50–59	13	11.1	37	31.6	67	57.3	117	100	
60–69	13	13.7	28	29.5	54	56.8	95	100	
70+	14	24.6	25	43.9	18	31.6	57	100	
Gender									0.001
Male	52	25.1	78	37.7	77	37.2	207	100	
Female	22	11.4	64	33.2	107	55.4	193	100	
Educational level									0.045
Illiterate	9	8.7	38	36.5	57	54.8	104	100	
Basic education	15	27.8	15	27.8	24	44.4	54	100	
Secondary education	7	18.4	12	31.6	19	50.0	38	100	
Diploma	11	25.0	14	31.8	19	43.2	44	100	
University graduate	19	27.1	27	38.6	24	34.3	70	100	
Post-graduate	13	14.4	36	40.0	41	45.6	90	100	
Marital status									0.010
Single	9	40.9	6	27.3	7	31.8	22	100	
Married	63	18.8	120	35.8	152	45.4	335	100	
Divorced/widow	2	4.7	16	37.2	25	58.1	43	100	
Work status									0.122
Not working	27	14.4	63	33.5	98	52.1	188	100	
Non-health care staff	20	19.6	44	43.1	38	37.3	102	100	
Health care staff	4	36.4	3	27.3	4	36.4	11	100	
Retired	23	23.2	32	32.3	44	44.4	99	100	
Monthly income									0.684
<5000 SR	28	15.9	59	33.5	89	50.6	176	100	
5001–10,000 SR	24	20.9	43	37.4	48	41.7	115	100	
10,001–15,000 SR	18	20.5	34	38.6	36	40.9	88	100	
>15,001 SR	4	19.0	6	28.6	11	52.4	21	100	
Co-morbidities									0.001
No	43	26.2	60	36.6	61	37.2	164	100	
Yes	31	13.1	82	34.7	123	52.1	236	100	
Smoking									0.125 ^
Non-smoker	59	16.8	129	36.6	164	46.6	352	100	
Ex-smoker	5	38.5	4	30.8	4	30.8	13	100	
Current smoker	10	28.6	9	25.7	16	45.7	35	100	
Made any attempts to control or reduce weight?									0.001
Yes	21	12.1	48	27.7	104	60.1	173	100	
No	53	23.3	94	41.4	80	35.2	227	100	

*p*: X^2^ test; ^: Likelihood Ratio.

**Table 3 healthcare-13-00658-t003:** Risk Factors of Obesity Among Type 2 Diabetic Participants in Abha, Saudi Arabia.

Risk Factors of Obesity	Total		Body Mass Index						*p*-Value
			Normal Weight		Overweight		Obese		
	No	%	No	%	No	%	No	%	
Duration of type 2 DM									0.017
<5 years	147	36.8	20	13.6	60	40.8	67	45.6	
5–9 years	97	24.3	15	15.5	30	30.9	52	53.6	
10–15 years	102	25.5	20	19.6	37	36.3	45	44.1	
>15 years	54	13.5	19	35.2	15	27.8	20	37.0	
Family history of obesity									0.019 ^
None	303	75.8	60	19.8	117	38.6	126	41.6	
One parent	59	14.8	10	16.9	17	28.8	32	54.2	
Both parents	38	9.5	4	10.5	8	21.1	26	68.4	
Do you eat breakfast regularly?									0.489
Yes	303	75.8	56	18.5	103	34.0	144	47.5	
No	97	24.3	18	18.6	39	40.2	40	41.2	
Do you eat lunch regularly?									0.440
Yes	333	83.3	65	19.5	115	34.5	153	45.9	
No	67	16.8	9	13.4	27	40.3	31	46.3	
Do you eat dinner regularly?									0.149
Yes	313	78.3	55	17.6	106	33.9	152	48.6	
No	87	21.8	19	21.8	36	41.4	32	36.8	
How many dyes per week eat the fruit?									0.137
<3 days/week	290	72.5	48	16.6	110	37.9	132	45.5	
>3 days/week	110	27.5	26	23.6	32	29.1	52	47.3	
How many dyes per week eat the vegetables?									0.523
<3 days/week	280	70.0	49	17.5	104	37.1	127	45.4	
>3 days/week	120	30.0	25	20.8	38	31.7	57	47.5	
How many days per week do you drink milk and cheese?									0.461
Never	149	37.3	25	16.8	53	35.6	71	47.7	
<3 days/week	147	36.8	26	17.7	48	32.7	73	49.7	
>3 days/week	104	26.0	23	22.1	41	39.4	40	38.5	
Have soft drinks									0.048
Yes	147	36.8	22	15.0	46	31.3	79	53.7	
No	253	63.3	52	20.6	96	37.9	105	41.5	
Frequency of having sweets/chocolate									0.477 ^
Never	240	60.0	41	17.1	84	35.0	115	47.9	
<3 days/week	128	32.0	29	22.7	47	36.7	52	40.6	
>3 days/week	32	8.0	4	12.5	11	34.4	17	53.1	

*p*: X^2^ test; ^: Likelihood Ratio.

**Table 4 healthcare-13-00658-t004:** Lifestyle Factors and Their Association with Body Mass Index (BMI) Among Type 2 Diabetic Participants in Abha, Saudi Arabia.

	Total		Body Mass Index						*p*-Value
			Normal Weight		Overweight		Obese		
	No	%	No	%	No	%	No	%	
Daily sleep hours (n = 400)									0.266 ^
<8 h	257	64.3	47	18.3	86	33.5	124	48.2	
8–10 h	136	34.0	24	17.6	55	40.4	57	41.9	
>10 h	7	1.8	3	42.9	1	14.3	3	42.9	
How many hours do you spend in front of the TV a day? (n = 400)									0.275
<1 h	113	28.3	27	23.9	39	34.5	47	41.6	
1–2 h	93	23.3	18	19.4	39	41.9	36	38.7	
2–4 h	129	32.3	18	14.0	44	34.1	67	51.9	
>4 h	65	16.3	11	16.9	20	30.8	34	52.3	
How many hours do you spend in front of the Mobile phone/computer a day? (n = 400)									0.150
<1 h	152	38.0	28	18.4	58	38.2	66	43.4	
1–2 h	69	17.3	13	18.8	25	36.2	31	44.9	
2–4 h	115	28.8	16	13.9	36	31.3	63	54.8	
>4 h	64	16.0	17	26.6	23	35.9	24	37.5	
Frequency of practicing exercises (n = 400)									0.020
Never exercise	116	29.0	15	12.9	36	31.0	65	56.0	
Irregularity	66	16.5	7	10.6	26	39.4	33	50.0	
Once or twice in week	89	22.3	17	19.1	36	40.4	36	40.4	
Many times, in weeks	64	16.0	14	21.9	23	35.9	27	42.2	
Daily	65	16.3	21	32.3	21	32.3	23	35.4	
If yes, type of exercise (n = 284)									-
Bike riding	2	0.7	1	50.0	0	0.0	1	50.0	
Power sports such as weightlifting	3	1.1	3	100.0	0	0.0	0	0.0	
Running	21	7.4	7	33.3	6	28.6	8	38.1	
Strength and fitness exercises	2	0.7	1	50.0	1	50.0	0	0.0	
Swimming	7	2.5	3	42.9	2	28.6	2	28.6	
Team sports such as football and basketball	4	1.4	2	50.0	2	50.0	0	0.0	
Walking	245	86.3	42	17.1	95	38.8	108	44.1	
Duration of exercise session/day (n = 284)									0.251 ^
<15 min	48	16.9	9	18.8	17	35.4	22	45.8	
15–30 min	146	51.4	28	19.2	56	38.4	62	42.5	
30–60 min	81	28.5	17	21.0	32	39.5	32	39.5	
>60 min	9	3.2	5	55.6	1	11.1	3	33.3	

*p*: X^2^ test; ^: Likelihood Ratio.

## Data Availability

The datasets used and/or analyzed in this study are available with the principal author upon reasonable request and will be provided by masking the identification of the individuals.

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
