# Peer review of "Prevalence and Risk Factors of Obesity Among Type 2 Diabetic Participants in Abha, Saudi Arabia: A Cross-Sectional Study"

_healthcare, 2025, doi:10.3390/healthcare13060658_

Round 1

Reviewer 1 Report

Comments and Suggestions for Authors

Thank you to the authors for giving me the opportunity to understand the obesity situation among diabetic patients in Abha, Saudi Arabia. Below are my review comments

Introduction

  1. On page 2, the statement that obesity is more prevalent among type 2 diabetes mellitus (T2DM) patients aligns with previous studies and common knowledge. This section should include a discussion of the gaps in prior research and how the current study aims to address these gaps.
  2. In this study, all BMI categories pertain to individuals with diabetes. The rationale for identifying obesity-related factors should be clearly stated. For example, is the focus on whether obese diabetic patients have poorer prognoses or more complications compared to normal-weight diabetic patients? This clarification should be added to the Introduction.

Methodology

  1. In section 2.1 Study Design and Settings, the study period is stated as "from November 2024 to January 2024," which appears to be incorrect. Please verify the correct time frame.
  2. The sample size of 400 participants should be justified by specifying the calculation method, the statistical software used, and its version.
  3. In section 2.3 Data Collection Tool and Techniques, the study utilized an online questionnaire, which may introduce selection bias, particularly among older adults or individuals with lower educational levels who may have difficulty completing the survey. Therefore, the statement "This approach aimed to achieve a broad and representative sample, ensuring the reliability and validity of the study findings" should be revised accordingly.
  4. The results indicate that 26% of participants were illiterate and 13.5% had only basic education. Given these figures, it is unclear how these participants were able to complete an online questionnaire. Please provide an explanation.

Results

  1. Percentages (%) in the tables should be removed.
  2. The total number of participants for some variables does not sum to 400 (e.g., gender: 399; smoking: 398). Please verify and ensure consistency across all variables. If missing values exist, they should be reported.
  3. The Monthly Income categories contain overlapping ranges (e.g., 5000–10,000 and 10,000–15,000). It is unclear which category should be selected for an income of exactly 10,000. Please clarify.
  4. In Tables 2, 3, and 4, BMI is categorized into three groups (normal weight, overweight, and obese) to examine group differences. However, some cells have n = 0 or n < 5. Meanwhile, Table 5 categorizes participants into only two groups (obese vs. non-obese). Please confirm whether the analysis should be consistently performed with two BMI groups across all tables for uniformity.
  5. In Table 5, obesity status is the dependent variable. However, the independent variables are not clearly defined in terms of their characteristics—whether they are categorical or continuous. Please specify.

Discussion

The discussion mainly compares the study results with existing diabetes literature, emphasizing similarities. However, it does not clearly highlight the study’s unique contributions. To strengthen this section:

  1. Highlight Novel Findings: Identify any unique trends or unexpected associations (e.g., between obesity and demographic factors like gender, education, or lifestyle) not emphasized in prior studies.
  2. Stratified Comparisons: If possible, analyze subgroups (e.g., age, gender, socioeconomic status) to provide a more detailed understanding of obesity risk factors in T2DM patients.
  3. Comparison with Non-Diabetic Populations: Referencing BMI and health behaviors in non-diabetic individuals could help contextualize the findings and strengthen arguments for targeted interventions.
  4. Clinical and Public Health Implications: Discuss how these findings inform obesity prevention and management strategies for diabetic patients in Saudi Arabia, considering culturally relevant factors like diet and physical activity etc.

Additional Suggestion:

While this study provides valuable insights into obesity among type 2 diabetic patients, a more comprehensive understanding of BMI differences and contributing factors could be achieved by including a comparison group of non-diabetic individuals. This would allow for a clearer assessment of whether the observed associations are specific to diabetic patients or part of broader population trends. Although incorporating such a comparison is beyond the scope of the current dataset, it may be worth considering for future research to enhance the study's impact.                    

Author Response

Reviewer 1

Introduction

  1. On page 2, the statement that obesity is more prevalent among type 2 diabetes mellitus (T2DM) patients aligns with previous studies and common knowledge. This section should include a discussion of the gaps in prior research and how the current study aims to address these gaps.

Reply: Thank you for your comment. Yes, the relationship between obesity and type 2 diabetes mellitus (T2DM) is well-established, but there are gaps in the literature regarding region-specific, dietary habits, physical activity patterns, and Sociocultural and Behavioral factors influencing obesity among diabetic patients, mainly in Saudi Arabia. Page 2 paragraph 4 lines 1 to 9.

  1. In this study, all BMI categories pertain to individuals with diabetes. The rationale for identifying obesity-related factors should be clearly stated. For example, is the focus on whether obese diabetic patients have poorer prognoses or more complications compared to normal-weight diabetic patients? This clarification should be added to the Introduction.

Reply: Thank you, the introduction has been modified and incorporated the Rationale for Studying Obesity in T2DM Patients. Page 2 paragraph 5 lines 1 to 11.

Methodology

  1. In section 2.1 Study Design and Settings, the study period is stated as "from November 2024 to January 2024," which appears to be incorrect. Please verify the correct time frame.

Reply: Thank you for pointing out the discrepancy in the study period, it was a typographical error instead of 15th January 2025 we typed January 2024. Page 3 2.1. Study Design and Settings paragraph 1 line 3.

  1. The sample size of 400 participants should be justified by specifying the calculation method, the statistical software used, and its version.

Reply: Thank you for your valuable comment, The sample size/ sampling section has been clearly explained.  Methodology section 2.2. sample and sampling section Page 3 in paragraph 2 lines 1-10.

  1. In section 2.3 Data Collection Tool and Techniques, the study utilized an online questionnaire, which may introduce selection bias, particularly among older adults or individuals with lower educational levels who may have difficulty completing the survey. Therefore, the statement "This approach aimed to achieve a broad and representative sample, ensuring the reliability and validity of the study findings" should be revised accordingly.

Reply: the methodology section has been revised and incorporated all the suggestions.   Methodology section 2.3 Data Collection Tool and Techniques section; Page 3 in paragraphs 4 &5.

  1. The results indicate that 26% of participants were illiterate and 13.5% had only basic education. Given these figures, it is unclear how these participants were able to complete an online questionnaire. Please provide an explanation.

Reply: It has been mentioned in the Data Collection Tools and Techniques part of Methodology section 2.3 Data Collection Tool and Techniques section; Page 3 in paragraph 5 lines 5 to 14.

Results

  1. Percentages (%) in the tables should be removed.

Reply: It has been removed and revised accordingly to enhance clarity and readability. In the results section all tables table 1-4.

  1. The total number of participants for some variables does not sum to 400 (e.g., gender: 399; smoking: 398). Please verify and ensure consistency across all variables. If missing values exist, they should be reported.

Reply: Thank you, we appreciate your eagle eye review, all the variables were reviewed, and the deficiencies. In the results section all tables table 1-4.

  1. The Monthly Income categories contain overlapping ranges (e.g., 5000–10,000 and 10,000–15,000). It is unclear which category should be selected for an income of exactly 10,000. Please clarify.

Reply: Thank you, for all the variables, reviewed and rectified, and improved clarity. In the results section all tables table 1-2.

  1. In Tables 2, 3, and 4, BMI is categorized into three groups (normal weight, overweight, and obese) to examine group differences. However, some cells have n = 0 or n < 5. Meanwhile.

Reply: It has been rectified, the p-value was estimated using the chi-square test, if the expected value of any cell is less than 5 in those cases we considered the Likelihood Ratio for the p-value. If the small cell value is “0” we have not considered calculating that variable. In the results section all tables table 1-4.

  1. Table 5 categorizes participants into only two groups (obese vs. non-obese). Please confirm whether the analysis should be consistently performed with two BMI groups across all tables for uniformity.

Reply: As per the advice of the statistician we deleted table 5. The Figure 2. Forest Plot of Risk Factors for Obesity Among Type 2 Diabetic Patients is also representing same, so statistician advised.

  1. In Table 5, obesity status is the dependent variable. However, the independent variables are not clearly defined in terms of their characteristics—whether they are categorical or continuous. Please specify.

Reply: As per the advice of the statistician we deleted table 5.

Discussion

The discussion mainly compares the study results with existing diabetes literature, emphasizing similarities. However, it does not clearly highlight the study’s unique contributions. To strengthen this section:

  1. Highlight Novel Findings: Identify any unique trends or unexpected associations (e.g., between obesity and demographic factors like gender, education, or lifestyle) not emphasized in prior studies.

Reply: It has been included in the discussion section, (e.g., gender, education level, and specific lifestyle habits) that have not been extensively studied in Saudi Arabia. In the discussion section Page 12 in paragraph 4 lines 1 to 14.

  1. Stratified Comparisons: If possible, analyze subgroups (e.g., age, gender, socioeconomic status) to provide a more detailed understanding of obesity risk factors in T2DM patients.

Reply: It has been included in the discussion section Page 11 paragraph 3 & 12 in paragraph 4 line 1 to 14.

  1. Comparison with Non-Diabetic Populations: Referencing BMI and health behaviors in non-diabetic individuals could help contextualize the findings and strengthen arguments for targeted interventions.

Reply: It has been included in the discussion section, this study's findings with obesity in the general Saudi population to provide contextual comparisons. in the discussion section Pages 12 & 13 in paragraph 5 lines 1 to 18.

  1. Clinical and Public Health Implications: Discuss how these findings inform obesity prevention and management strategies for diabetic patients in Saudi Arabia, considering culturally relevant factors like diet and physical activity, etc.

Reply: It has been included in the discussion section in the discussion section Page 13 paragraph 2 lines 1 to 14.

Additional Suggestion:

While this study provides valuable insights into obesity among type 2 diabetic patients, a more comprehensive understanding of BMI differences and contributing factors could be achieved by including a comparison group of non-diabetic individuals. This would allow for a clearer assessment of whether the observed associations are specific to diabetic patients or part of broader population trends. Although incorporating such a comparison is beyond the scope of the current dataset, it may be worth considering for future research to enhance the study's impact.

Reply: Thank you for your excellent and thought-provoking suggestion, that comparing our findings with non-diabetic populations could strengthen our arguments. While our current dataset does not include non-diabetic participants, we will plan further studies to assess whether the observed obesity associations are specific to diabetic patients or part of broader population trends. in the limitations section Page 13 paragraph 3 lines 1 to 11.

Reviewer 2 Report

Comments and Suggestions for Authors

I would like to thank the editor for the opportunity to review this study. The study appears to present relevant results and is in agreement with similar results in other areas, presenting a serious public health problem worldwide, but it does not contribute anything new with respect to other studies but rather corroborates what already exists. However, I believe that the study, at its core, presents many deficiencies in the analysis and I believe that they will not be able to solve them, so my recommendation is to reject the article, since I believe that the errors committed will not be solved with a major revision. Below I justify my decision.

The idea of ​​the study is a good idea, on a current topic, studying a problem that concerns any government of any country in the world.

The main problem is that a non-probabilistic type of sampling, such as snowball sampling, has been used to select the sample, which means that not all individuals in the population have the same probability of being selected. This differentiates it from probability sampling methods and limits the ability to generalize the results. Non-random sampling techniques lead researchers to gather what are commonly known as convenience samples, thus providing results that are difficult to generalize beyond the sample studied, in addition to the little control over the sampling method. The subjects that the researcher can obtain are mainly based on previously observed subjects. This oversampling of a particular peer network can lead to a possible bias. Therefore, the results may not be generalizable to the entire population due to the non-probabilistic nature of the method. It is usually impossible to determine the sampling error or make inferences about the populations from the sample obtained. Normally, snowball sampling is a valuable technique to access hard-to-reach populations and conduct exploratory studies. However, it is crucial to be aware of its limitations and potential biases when interpreting the results. In this case, I find it difficult to understand why they have used this type of sampling, since the people diagnosed with this disease will be registered, either in health centers or in hospitals, so a list of them can be kept and the population under study within the region can be determined, a figure that they do not give, but which can be obtained, if not for all, at least approximately, for the majority, since the patient diagnosed with T2DM has been in a hospital center and there is a record of it in that region. This will be important to determine the sample size, another important problem.

Snowball sampling offers several advantages, such as access to hard-to-reach populations (which is not the case) and efficiency in terms of costs and time. However, it is essential to consider the limitations, including sampling bias and lack of control over the composition of the sample. Researchers should carefully weigh these pros and cons when deciding whether to use snowball sampling in their studies, ensuring that the chosen method aligns with the research objectives and target population.

As I said before, another problem concerns the sample size, since a size of 400 can be considered small, because although we do not know the population under study, or at least they do not indicate it, although I believe, as I said, that it could be known, in a region of almost 350,000 inhabitants, according to data from 2022, they do not indicate how they determined the optimal sample size, but rather that they are the people who were able to connect to the internet and answered the survey provided. In the work, the authors limit themselves to saying "was targeted to ensure sufficient statistical power for detecting significant associations", but they do not say what power they have, how they estimated it, with what software, since the software used does not estimate it, and it is because they cannot, because the type II error is uncontrolled and they cannot guarantee that power that they say they achieve or claim to justify.

Another serious problem appears in section 2.3 when the authors say that: "Data collection was done using an online questionnaire designed based on an extensive literature review of previous similar studies, ensuring its reliability and validity". They have made a questionnaire based on their experience with a review of similar studies, but in studies of this type either statistically validated questionnaires used and validated by other authors are used, which allows comparisons with them, or it has to be validated, but your experience does not guarantee statistical validity, validity that is useful for your use, but without being statistically validated, the results obtained are not valid. That is, at no time do they say that they have validated it, what instrument they have used to validate it or anything, which makes us doubt even more that the data obtained are valid, and therefore the results and conclusions, although they may be similar to other studies carried out, since the conditions of the population studied cannot be similar to other studies carried out, which they do not talk about either.

Based on this online questionnaire that the respondents provided, we could detect more errors and biases, such as measurement bias. To give a simple example. Each person has provided their weight in kg, but each one will have used a different scale that they had at home, of a different brand, with a different precision and that can provide measurement errors between the different people surveyed. Not to mention, for example, with the height in meters, which leads us to another variable such as the body mass index. I have only mentioned 3 variables, but what I want to give as an example is that there can be many differences from one data to another just due to measurement errors from not using the same measuring device or doing it in the same way, something that increases the number of basic errors in the work.

Finally, although the authors mention some of the limitations of the work, this does not exempt them from the errors committed, so I consider that the work has some basic errors that are difficult to resolve.

Author Response

Respected Reviewer,

The Authors sincerely appreciate the Reviewer's time and effort in reviewing our manuscript entitled “Prevalence and Risk Factors of Obesity Among Type 2 Diabetic Patients in Abha, Saudi Arabia: A Cross-Sectional Study.” Your constructive feedback has been invaluable in refining our study. The authors incorporated the reviewers’ suggestions/concerns and revised the manuscript accordingly.

Reviewer 2

2. Questions for General Evaluation

Reviewer’s Evaluation

Response and Revisions

Does the introduction provide sufficient background and include all relevant references?

Yes/ can be improved

Yes, it has been revised and incorporated the novelty and rationale for the study

Are all the cited references relevant to the research?

Yes

The references were relevant to the study

Is the research design appropriate?

Must be improved

The Methodology was completely revised and rewritten

Are the methods adequately described?

Yes/Must be improved

The Methodology was completely revised and rewritten

Are the results clearly presented?

Yes/Can be improved

It has been revised

Are the conclusions supported by the results?

-

It has been revised

Comment: 1. The study does not contribute anything new compared to previous studies but rather corroborates existing knowledge.

Reply: We acknowledge that our study confirms many previously established findings regarding obesity among type 2 diabetic patients. However, our research contributes new insights by providing region-specific data for Abha, Saudi Arabia, we also examine sociocultural and behavioral factors unique to the Saudi population, including soft drink consumption, meal patterns, and gender disparities in physical activity levels. An area where limited research on this topic exists. Unlike many Western studies, these aspects have not been extensively explored in prior studies from the region. In the introduction, results, and discussion section it was highlighted.

Comment: 2. The use of snowball sampling limits the generalizability of results. Since diabetic patients are registered in health centers and hospitals, a probability-based sampling method should have been used.

Reply: We recognize the limitations of snowball sampling and acknowledge that probability sampling would provide greater generalizability. However, due to data access restrictions, obtaining a full list of diabetic patients from health centers was not feasible. To overcome this challenge, we employed a hybrid approach: Initial recruitment from health centers – We obtained a list of diabetic patients with contact details from local health centers and reached out to them directly. Participant-driven recruitment – Patients were encouraged to share the survey with their peers, ensuring a diverse representation of the diabetic population in Abha.  Methodology section 2.2. sample and sampling section Page 3 in paragraph 2 lines 1-10.

To minimize the bias, the authors collected a relatively adequate sample (400 participants), above the calculated minimum sample size of 356. And ensured diverse representation in terms of age, gender, socioeconomic status, and education level. Methodology section 2.2. sample and sampling section Page 3 in paragraph 2 lines 1-10.

Comment: 3. The manuscript does not explain how the required sample size was determined or what power analysis was conducted.

Reply: The Methodology section has been revised and incorporated as much as possible information. The minimum sample size was calculated using an online sample size calculator at a 95% confidence level, a 5% margin of error, and an estimated obesity prevalence of 30% based on previous studies. The estimated sample size was 323, with an additional 10% added for non-response bias, resulting in a minimum required sample of 356. The final sample included 400 participants, ensuring sufficient statistical power for detecting significant associations.  Methodology section 2.2. sample and sampling section Page 3 in paragraph 2 lines 1-10.

Comment: 4. The study does not state whether the questionnaire was statistically validated. The use of self-reported weight and height may introduce measurement errors.

Reply:  It has been revised and now explicitly described our questionnaire validation process: The questionnaire content validity was reviewed by two independent public health experts to ensure comprehensive coverage of relevant factors. A pilot Test was Conducted with 25 participants, leading to refinements in clarity and cultural appropriateness. Cronbach’s alpha was calculated at 0.78, indicating acceptable internal consistency. The questionnaire was translated into Arabic and back-translated into English by two independent translators to ensure accuracy.

Regarding measurement bias, we acknowledge that self-reported anthropometric data (weight, height) may introduce variability. To reduce this bias: We instructed participants to use recent measurements taken within the past 30 days. We conducted sensitivity analyses to assess whether extreme values influenced the results. Methodology section 2.3 Data Collection Tool and Techniques section; Page 3 in paragraphs 4 &5.

Comment: 5. Different scales and measuring devices used by participants at home could introduce errors in BMI calculation.

Reply: We accept that variation in home sales may affect BMI accuracy. However, to minimize measurement errors, the authors requested that participants report their most recent clinical measurements which were done at health center follow-up visits. to reduce inconsistencies in BMI calculations excluded extreme outliers that suggested implausible values. Methodology section 2.3 Data Collection Tool and Techniques section; Page 4 in paragraph 1 section 2 of questionnaire.

Comment: 6. The authors acknowledge some limitations, but these do not fully compensate for the study’s fundamental errors.

Reply: We agree that all researches have limitations, and we have expanded the Limitations section to address key concerns: The use of snowball sampling limits the generalizability of findings. Future studies should explore randomized or clinic-based sampling. Measurement errors in weight and height may have influenced BMI calculations. We acknowledge this and suggest incorporating clinical measurements in future research. This cross-sectional study cannot establish causality. Longitudinal studies are needed to confirm causal relationships between obesity risk factors and T2DM progression. in the limitations section Page 13 paragraph 3 lines 1 to 11.

We acknowledge the limitations of our study, we firmly believe that our research provides valuable insights into the prevalence and risk factors of obesity among T2DM patients in Abha, Saudi Arabia. Our study helps bridge the gap in region-specific research and highlights modifiable lifestyle factors that can inform targeted public health interventions.

The authors sincerely appreciate the reviewer’s detailed feedback, which has allowed us to strengthen the manuscript and present a more robust and transparent study. We hope that the revisions sufficiently address the concerns raised and demonstrate the value of our research.

Thank you for your kind consideration.

Sincerely,
Dr Bayapa Reddy N.

Round 2

Reviewer 1 Report

Comments and Suggestions for Authors

The researchers have addressed most of the comments; however, some issues remain:

1.Some variables' total percentages exceed 100%, such as in Table 1 (age, marital status, employment, income) and Figure 1. Please carefully review all tables and figures to ensure accuracy in numerical values.

2.Diabetes has been extensively studied, yet the authors have cited numerous outdated references. Please incorporate more recent literature (preferably within the last five years) to support this study.

3.Since the p-values are already presented in the tables and clearly indicate whether they are below 0.05, please remove the asterisks (*) and the "p < 0.05" notation.

Author Response

Respected Reviewer,

The Authors sincerely appreciate the Editorial board time and effort in reviewing our manuscript entitled “Prevalence and Risk Factors of Obesity Among Type 2 Diabetic Patients in Abha, Saudi Arabia: A Cross-Sectional Study.” Your constructive feedback has been invaluable in refining our study. The authors incorporated the reviewers’ suggestions/concerns and revised the manuscript accordingly.

Reviewer 1                                                                 Round 2

2. Questions for General Evaluation

Reviewer’s Evaluation

Response and Revisions

Does the introduction provide sufficient background and include all relevant references?

Can be improved

Yes, it has been revised and incorporated the recent references and improved.

Are all the cited references relevant to the research?

Can be improved

The references were updated with intensive search, the authors tried their best to incorporate the last five years, relevant references to the study

Is the research design appropriate?

Can be improved

The Study Design and Settings revised

Are the methods adequately described?

Yes

-

Are the results clearly presented?

Can be improved

It has been revised all the tables and graphs

Are the conclusions supported by the results?

Yes

-

Comment: 1. Some variables' total percentages exceed 100%, such as in Table 1 (age, marital status, employment, income) and Figure 1. Please carefully review all tables and figures to ensure accuracy in numerical values.

Reply: Thank you for highlighting the very important issue, we have completely revised all the tables and the results section.

  1. Diabetes has been extensively studied, yet the authors have cited numerous outdated references. Please incorporate more recent literature (preferably within the last five years) to support this study.

Reply: The references were updated with an intensive search, the authors tried their best to incorporate the last five years relevant references to this study, around 20% of the references were very relevant to the study, which were a little older, but those references were also within the 10 years period.

  1. Since the p-values are already presented in the tables and indicate whether they are below 0.05, please remove the asterisks (*) and the "p < 0.05" notation.

Reply: It was deleted as per the suggestion.

The authors sincerely appreciate the reviewer’s detailed feedback, which has allowed us to strengthen the manuscript and present a more robust and transparent study. We hope that the revisions sufficiently address the concerns raised and demonstrate the value of our research.

Thank you for your kind consideration.

Sincerely,
Dr Bayapa Reddy N.